# Perceptions of barriers and facilitators to opioid reduction after total joint arthroplasty among orthopedic surgeons practicing in Canada, Japan, and the Netherlands: A qualitative description study

Mansi Patel[1], Parsia Parnian[2], Kim Madden[1,3], Sheila Sprague[1,3], Anita Acai[4,5,6], Ydo Kleinlugtenbelt[7], Natsumi Saka[8], Ellie Landman[7], Harsha Shanthanna[3], Vickas Khanna[1], Jason W. Busse[3,9]*

1 Division of Orthopaedics, Department of Surgery, McMaster University, Hamilton, ON, Canada, 2 Faculty of Health Sciences, McMaster University, Hamilton, ON, Canada, 3 Department of Health Research Methods, Evidence, and Impact, McMaster University, Hamilton, ON, Canada, 4 Department of Psychiatry & Behavioural Neurosciences, McMaster University, Hamilton, ON, Canada, 5 McMaster Education Research, Innovation and Theory (MERIT) Program, McMaster University, Hamilton, ON, Canada, 6 St. Joseph's Education Research Centre (SERC), St. Joseph's Healthcare Hamilton, Hamilton, ON, Canada, 7 Department of Orthopedic and Trauma Surgery, Deventer Ziekenhuis, Deventer, Netherlands, 8 Department of Orthopedic Surgery, Teikyo University School of Medicine, Tokyo, Japan, 9 Department of Anesthesia, McMaster University, Hamilton, ON, Canada

* bussejw@mcmaster.ca

## Abstract

Opioid analgesics are commonly prescribed after total knee and hip arthroplasty to manage pain. Rates of opioid prescribing after arthroplasty differ by country, suggesting differences in policies or surgeons' practices. We adopted a qualitative description design to explore and compare Canadian, Dutch, and Japanese orthopaedic surgeons' perceptions of facilitators and barriers to opioid reduction after total joint arthroplasty. We used a combination of convenience and purposive sampling, and snowball recruitment to facilitate 27 semi-structured interviews online or via a phone call. We concurrently collected and analyzed data using conventional (inductive) content analysis. In our sample, all Canadian surgeons and almost all Dutch surgeons prescribed opioids to all arthroplasty patients post-discharge. Surgeons in Japan showed much greater variability, with half of those interviewed prescribing opioids to only a minority or no patients post-discharge. Japanese surgeons indicated that a 10–30-day hospital stay was typical after surgery and believed that opioids were often unnecessary for managing postoperative pain. Dutch surgeons described using an institutional standard pain management protocol, while Canadian and Japanese surgeons noted high variability in the type and dose of opioids prescribed, even within the same institution. Orthopaedic surgeons in each country identified challenges and facilitators to reduced postoperative opioid use in six key areas: (1) opioid prescribing practices, (2) patient factors,

**Data availability statement:** To protect the privacy or confidentiality of human research participants, not all data can be made available publicly (i.e. direct quotations of participants that may reveal their identity). Interested researchers can contact the corresponding author or McMaster University Office of Scholarly Communication at scom@mcmaster.ca to access the minimal data that we have. They can also access this data at https://mac-sphere.mcmaster.ca/handle/11375/31823.

**Funding:** One of the authors (MP) has received funding from the Mitacs Globalink Research Award for this study. The funders did not play any role in the study design, data collection and analysis, decision to publish, or preparation of the manuscript.

**Competing interests:** I have read the journal's policy and the authors of this manuscript have the following competing interests: One of the authors (MP) has received funding from the Mitacs Globalink Research Award. One of the authors (VK) is a consultant for Stryker Canada and Zimmer Biomet Canada, and has a spouse who works for Stryker Canada. One of the authors (YK) received a grant from Pioneers in Healthcare, received honoraria from Agnovos, received support for meeting attendance from RGS, and payment from NOV for participation in a Data Safety Monitoring Board or Advisory Board. One of the authors (KM) received a grant from CIHR (BioTalent), is a participant in a Data Safety Monitoring Board or Advisory Board at the University of Calgary, and has a leadership or fiduciary role at the Clinical Orthopaedics and Related Research and Canadian Orthopaedic Association. All other authors (PP, SS, AA, NS, EL, HS, JWB) certify that there are no funding or commercial associations (consultancies, stock ownership, equity interest, patent/licensing arrangements, etc.) that might pose a conflict of interest in connection with the submitted article related to the author or any immediate family members.

(3) collaborative care, (4) opioid prescribing policies/guidelines, (5) surgeon education, and (6) personal perceptions/beliefs. Canadian, Dutch, and Japanese orthopedic surgeons in our study described a range of individual, patient, and system level contributors to variability in opioid prescribing after joint replacement surgery. These findings suggest that multifactorial and context-specific approaches may be required to address barriers and optimize postoperative use of opioids.

## Introduction

The prevalence of osteoarthritis (OA) is increasing, and total joint arthroplasty (TJA) is a commonly used procedure to address advanced stages of the disease. However, approximately 29% of patients report experiencing moderate to severe acute pain after total hip arthroplasty (THA) and 51% after total knee arthroplasty (TKA) [1]. Consequently, opioids are often prescribed in the acute postoperative period after joint replacement surgery in many countries [2]. However, the increase in opioid prescribing has contributed to the current opioid crisis, referring to the increase in the incidence of deaths and hospitalizations resulting from the rising use of prescription and non-prescription opioids [3]. Variations in opioid prescribing and consumption across countries suggest disparities in healthcare systems, policies, and surgeon perspectives [4–6].

### Canada

North America is at the heart of the opioid crisis, with Canada and the USA leading in the per-person consumption of prescription opioids globally [7]. The opioid crisis primarily emerged due to the insufficient regulation and profit-driven motives of the pharmaceutical and healthcare industries, which nearly quadrupled opioid prescribing [7]. There were 22,828 deaths from opioid toxicity between 2016 and 2021 in Canada, with 6,946 deaths occurring during the COVID-19 pandemic between April 2020 to March 2021 [8].

### The Netherlands

In the Netherlands, between 2008 and 2017, the cumulative number of opioid prescriptions nearly doubled, with a concordant increase in misuse [9]. Between 2007 and 2016, deaths caused by opioid overdose in Europe were 1.2 deaths per 100000 per year, compared to 15 deaths per 100000 in the USA [10]. Opioid misuse in the Netherlands is substantially lower than in North America, including opioid-related mortality, though it is much greater than in Japan [9,11]. Concerns about opioid use in the Netherlands are predominantly due to marketing by pharmaceutical companies directed to the surgeon, and surgeon preferences [9].

### Japan

The per capita use of six potent opioids is 26 times higher in the USA than in Japan despite a similar prevalence of chronic pain. Orthopaedic surgeons in Japan prescribe significantly fewer postoperative opioids compared to North America [4]. Japan

was the first Asian country to enact modern legislation in the 19th century to regulate opioid consumption strictly [4,12]. Therefore, in the last couple of decades, Japan has not experienced substantial issues with opioid addiction and related consequences [4].

### Rationale

Comparing and understanding perceptions of facilitators and barriers to reducing the use of opioids in distinct jurisdictions may inform the willingness of surgeons to revisit their prescribing practices. It may also aid in identifying potential best practices that can be shared among countries to enhance patient outcomes and promote safer pain management practices. We therefore conducted a qualitative study to explore: (1) perceptions of opioid prescribing after joint replacement, and (2) facilitators and barriers to opioid reduction after TKA or THA in three jurisdictions that represent high (Canada), moderate (The Netherlands), and low (Japan) opioid prescribing practices.

## Methods

### Research design

We used a qualitative description design, commonly employed to provide descriptive accounts of health care phenomena. Qualitative description methodology emphasizes an analytical and data-driven perspective to prioritize the practical implications of findings for healthcare providers, educators, and policymakers [13–16].

### Sampling and recruitment

Surgeons from Canada, the Netherlands, and Japan were sampled. We recruited surgeons who understood and spoke English and routinely performed hip or knee arthroplasty via convenience sampling. This entailed contacting surgeons within our personal and professional networks who we knew either prescribed or did not prescribe opioids. We used non-probability, purposive sampling to intentionally "select" participants who could provide a rich account of the phenomena of interest [16,17]. We also used snowball recruitment, where initial participants recommended additional contacts in the same country who fit the research criteria and were willing to partake in the study. We sought maximum variation in academic versus community practice locations, gender, and years of practice. We also recruited participants via our professional social media accounts (i.e., Instagram and Twitter) and email.

### Interview guide

We used a semi-structured interview guide, which consisted of a series of open-ended questions to capture the perceptions of surgeons on barriers and facilitators to reduced opioid prescribing after discharge following TKA or THA (S1 File). The research objectives, the team members' clinical experiences and perceptions of important topics, and past literature concomitantly informed the development of this interview guide [18,19].

### Data collection

Trained research assistants conducted semi-structured interviews via Zoom or phone with 11 Canadian, 9 Dutch, and 7 Japanese orthopaedic surgeons. One interview with a Dutch surgeon included an anesthesiologist as well. We provided Japanese surgeons with an option to access a translator if needed. Data saturation was reached after no new themes were developed from data analysis, and we completed two subsequent interviews in each country to verify [16]. Interviews were completed between November 2022 and February 2023, lasting between 15–45 minutes. Each interview was audio-recorded, transcribed verbatim, and de-identified for data analysis. Prior to data collection, participants were provided with a research ethics board (REB)-approved information letter that described the study in accessible language and provided sufficient information for participants to make an informed decision about their participation. They were

given ample time to review and ask questions. Thereupon, verbal informed consent was obtained during the interview and documented on a REB-approved verbal informed consent log for audio and/or video recording and the use of quotes. This study was reviewed and approved by the Hamilton Integrated Research Ethics Board (Project number 15023; McMaster University).

### Data analysis

Data analysis involved conventional (inductive) content analysis [14,20,21]. We collected and analyzed data concurrently. Two research team members iteratively read the transcripts to derive codes by highlighting text that captured key thoughts or concepts. Subsequently, labels for codes were actively constructed from the text and formed the initial coding scheme. We grouped the codes into categories based on their relationships and connections, which helped organize codes into meaningful groups [20]. To ensure the authenticity and completeness of the identified concepts and their overarching categories, two members coded all transcripts independently and in duplicate in a triangulated approach. We also captured representative quotes for the identified categories.

To compare themes in each country, we used an integrated, joint, side-by-side display [22]. We illustrated concepts in Venn diagrams to identify similarities and differences between jurisdictions. We summarized demographic data using ranges and proportions. We used Dedoose (v9.0.90) for data management and analysis.

### Strategies to promote rigor and trustworthiness

The term "bias" is generally drawn from quantitative research and is incompatible with the philosophical underpinnings of qualitative methodology as the subjectivity resulting from the researcher is integral to qualitative research [15]. Accordingly, to enhance the credibility of the data, the research team comprised expert qualitative researchers, orthopaedic surgeons, an experienced orthopaedic research coordinator, and experts in chronic pain management. The team had peer debriefing meetings to ensure conceptual ideas were explored and well-organized. We also used triangulation to enhance the trustworthiness of our findings [23]. We used member checking, where we asked participants to check the completeness of the summary of findings based on their responses at the end of the interview. The interviewers also developed positionality statements (S2 File), and kept a field diary and a reflective journal to ensure interpretive authority [24]. Potential power imbalances were minimized as interviews were conducted by graduate and undergraduate students with no authority or influence over participants' careers.

### Inclusivity in global research

Additional information regarding the ethical, cultural, and scientific considerations specific to inclusivity in global research is included in the Supporting Information (S4 File).

## Results

### Characteristics of participants

Out of the 27 participants in the study, the majority were men (n = 21, 78%) (Table 1). The length of surgical practice varied, ranging from less than one year to 28 years. Sixty-three percent of included surgeons practiced in academic settings. In-hospital stay post-operation was 0–3 days for both TKA and THA in Canada and the Netherlands, and 10–30 days in Japan. Interviewed surgeons in Canada and the Netherlands reported that all or almost all their patients were prescribed opioids after discharge. In contrast, surgeons in Japan showed much greater variability, with half prescribing opioids to all patients and the other half prescribing opioids to only a minority or no patients after discharge. In addition, during interviews, Canadian and Dutch orthopaedic surgeons indicated use of a wider range of opioids as compared to Japanese surgeons, who predominantly prescribed NSAIDs and tramadol (S3 File).

**Table 1. Demographic and practice characteristics of participating orthopaedic surgeons in Canada, The Netherlands, and Japan.**

| | Canada (N = 11) | The Netherlands (N = 9) | Japan (N = 7) | Overall (N = 27) |
|---|---|---|---|---|
| **Male** | 9 (82) | 6 (67) | 6 (86) | 21 (78) |
| **Years of practise** | <1–20 | 2–20 | 11–28 | <1–28 |
| **Practise setting**[a] | | | | |
| Academic | 7 (64) | 7 (78) | 3 (50) | 17 (63) |
| Community | 3 (27) | 2 (22) | 0 | 5 (19) |
| Both | 1 (9) | 0 | 3 (50) | 4 (15) |
| **Arthroplasty Procedures Performed Annually** | 105–600 | 50–600 | 24–220 | |
| TKA | 70–300 | 30–300 | 0–150 | |
| THA | 15–300 | 0–300 | 0–150 | |
| **Average LOS (days)** | | | | |
| TKA | 0–3 | 0–3 | 10–30 | |
| THA | 0–3 | 0–3 | 10–30 | |

[a]N = 6 for Japan (one participant did not answer the question); TKA, Total Knee Arthroplasty; THA, Total Hip Arthroplasty; LOS, Length of Stay.

We grouped data on barriers and facilitators to opioid reduction into six categories (S3 File, S1 and S2 Figs). Examples of quotes representing each theme can be found in Table 2.

### Category 1: Surgeons' prescribing practices

We identified several facilitators to opioid reduction emerging from surgeons' interview responses (S3 File, S1 and S2 Figs). Canadian and Dutch surgeons included in our study stated that they had decreased the quantity and duration of opioid prescriptions over time, with many currently prescribing opioids as-needed. Moreover, during interviews, Canadian and Dutch surgeons shared concerns about misuse and the adverse effects of opioids, specifically addiction. Conversely, Japanese surgeons expressed apprehension about additional side effects, such as falls, sedation, nausea, constipation, and hepatic dysfunction, in addition to addiction. Many Japanese surgeons interviewed in this study believed that opioids were not necessary for pain management after discharge and preferred to use non-opioid alternatives such as NSAIDs or paracetamol. However, half of the Japanese surgeons we interviewed prescribed tramadol to all their patients on discharge. Meanwhile, in the Netherlands, all included surgeons described using a standard institutional protocol for postoperative/discharge pain management, which reduced variations in prescribing practices among colleagues.

Participating surgeons in all three countries reported encountering several barriers and challenges to opioid reduction. Dutch and Canadian surgeons highlighted the tension they faced between the need to relieve patients' pain and preserve their mobility while minimizing the risks of opioid use. These surgeons also noted that busy schedules hindered their ability to monitor patients' opioid use. Although many Dutch surgeons who were interviewed followed standard institutional protocols for postoperative and discharge prescribing, some surgeons were unfamiliar with the specific details because anesthesiologists handled postoperative pain management at some institutions. However, most Canadian and Japanese orthopaedic surgeons in our study did not describe using a standard institutional protocol. Many Canadian surgeons perceived that senior orthopaedic surgeon exhibited resistance to reducing opioids and had outdated prescribing practices. They also pointed out that differences in individual surgeons' preferences, experiences, and training may explain variations in prescribing practices.

### Category 2: Patient factors

All Dutch and Canadian orthopaedic surgeons interviewed in our study reported that either they or a member of the care team provided patients with pre- and postoperative education about pain expectations, medications, and potential side

**Table 2. Categories of barriers and facilitators to reduced opioid prescribing identified by surgeons.**

| Category | Example Quotes |
|---|---|
| 1. Surgeons' opioid prescribing practices | "I think the risk of addictive potential outweighs the benefit of you not having the amount of pain that you have." – Participant 1855, CA<br>"I recommend that don't use opioids. We can manage without opioids for postoperative patients. That is my message." – Participant 2817, JAP<br>"Finding the right balance between pain management, pain relief, on one side, and on the other side, trying to avoid addiction, trying to avoid side effects. Yeah, I think those are two most important things. So, it's always a struggle because on the one hand, you want to have your patients to mobilize early and quickly because that will provide them a better outcome on a functional level. On the other hand, you cannot achieve that without any pain medications. But you don't want to give them too much pain medication, and you don't want all the negative side effects of the pain medication. So, it's finding the optimal balance in between them." – Participant 1258, NL<br>"I have a guy one year in, and I have a guy there for 25 or 30 years. So, we're creatures of habit. It's not an easy thing to break what we've been doing for many, many years. So, newer medications, or at least medications we use more commonly to date, are more likely be used by a newer surgeon, than somebody that's been practicing for 20 to 25 to 30 years." – Participant 2116, CA |
| 2. Patient factors | "My patients don't want to prescribe opioids, because Japanese are resistant to opioids…" – Participant 2756, JAP<br>"I think that for a while, we had this concept of like pain free surgery that somehow, we were going to magically have people have big operations and not have any hint of pain. Well, the only way to do that is to stay unconscious until you're healed, right?" – Participant 1205, CA<br>"Well, I don't know if it counts, but I always say there is no surgery without pain. So, to explain to [patients] they can expect and will have some pain, but we will try to make it manageable, which is not the same as having no pain at all. So, that's the counsel, the consultation I give them beforehand. Afterwards, we do have the pain scale, and we can try and teach them, well, this is what you can expect. This is normal. This will reside within a few days so, they know what to expect." – Participant 2491, NL |
| 3. Collaborative care | "So, the tricky part is that it's not always my decision. So, I would love it to be something that I'm always involved in. But patients sometimes go to their family doctors and then go to other prescribers, right? And we do have some information that, you know, a lot of the prescriptions within the first sort of 6 weeks after surgery are by others. And they don't necessarily inform us when they're prescribing a patient more pain medication, right." – Participant 1205, CA<br>"So once a year, we have a discussion with our anesthesiologists, nurse practitioners, and orthopedic surgeons about treatments for our arthroplasty patients. And, that also includes pain medications, and it also includes how and what we should prescribe and for how long. So, once a year we discuss it, and then two or three of our arthroplasty surgeons attend that meeting, and then it is discussed for everybody." – Participant 2812, NL<br>"Unfortunately, I don't have the pain management team in my institute." – Participant 1260, JAP<br>"In addition, in general, in Japan, patients move from teaching hospital to rehab facility after two weeks so he doesn't know what patients take 2 weeks post-operatively until 3 months post-operatively." – Participant 2066, JAP |
| 4. Policies, guidelines, and regulations on opioid prescribing | "I think they're driven, mainly by the opioid crisis in the States. So, like surgeons down there can go to prison for example if they're found to be overprescribing but that may not be intentional so there's a reluctance to prescribe in today's climate." – Participant 1855, CA<br>"I think one of the challenges we run into is that the hospital, you know, has developed these policies about how many opioids you should prescribe to people, and the residents get a lot of pressure in our electronic system about how many tablets can be on a prescription. And whilst that's very good, I think there are some things that are just very painful and that require more than, you know, 20 tabs of an opioid. Like if you have your knee replaced, odds are, you're going to need more than 20 tablets or something. So, I think that's a struggle because I think, again, there are some patients who need more than what the hospital policies are designed for." – Participant 1983, CA<br>"So, we think that there must be some opioid using guideline used among anesthesiologists. But, we, orthopedic surgeons, do not know the detail about the guideline." – Participant 2066, JAP<br>"Good question, wouldn't know, to be honest, and there certainly will be [guidelines]. But I've never used them." – Participant 2662, NL |
| 5. Opioid risk mitigation via surgeon education | "Well, there's always a change. And like I said, in the past, I used to prescribe tramadol or oxycodone, and now its morphine. So, things develop over time. So, it's good to have an update on what's new in the market. What's new, what are the new insights, study results, which opioids, which pain medication is the best to use on what type of patient, for how long." – Participant 1258, NL<br>"I think that young doctors need education for giving opioids." – Participant 2066, JAP<br>"First of all, drug company gives the seminar. A small seminar to use opioids. So, I learn from that." – Participant 2297, JAP<br>"No, I just think that continuing medical education is an excellent concept, and something that it probably should almost be required for us if we are prescribing these drugs, which I consider to be pretty dangerous, you know. And so, I think that it would be certainly a reasonable thing and it would, you know, benefit us and our patients." – Participant 2658, CA |

*(Continued)*

**Table 2.** (Continued)

| Category | Example Quotes |
|---|---|
| 6. Personal perceptions and beliefs on opioid prescribing | "And, yeah, I mean it is an arthroplasty procedure, it does hurt, to some degree, so I don't think that completely eliminating narcotics is necessary either. We just have to be very cognizant of dependence and abuse, and be very restrictive on how we do that." – Participant 2373, CA<br>"There are still patients who will suffer pain. So, we want to use stronger painkiller. As mentioned before, the amount of the opioid we can use in Japan is very restricted, so we can use very small amount. So, we want to use more, but we cannot." – Participant 2297, JAP<br>"I think we have a strong challenge ahead of us. So, we're still working on what are the ideal ways to further reduce opioid use. So, I think for what's available, it's our best option right now. But I think we have to always keep in mind there's room for improvement. Which being said, I don't think it's necessary to avoid opioids at all. But I think you should make maximum efforts to reduce complications caused by opioid use and mainly dependency, of course." – Participant 2632, NL |

effects (S3 File, S1 and S2 Figs). Dutch surgeons also described offering education about the pain management ladder, which prioritizes non-opioid analgesics first and strategies for tapering opioid use. Some Japanese surgeons did not provide patient education regarding analgesia as they believed it was only necessary when prescribing strong opioids. Orthopaedic surgeons in Japan noted that cultural factors led to resistance to opioid use among some Japanese patients, who preferred non-opioid pain management options.

Canadian and Dutch participants shared concerns about the potential for patients to divert unused opioids; however, most surgeons reported that they did not provide instructions to patients about how to properly dispose of unused opioids. Surgeons in both countries expressed concerns about patient dissatisfaction with reduced quantity/dosage of opioids and the use of non-opioid pain management modalities. Canadian and Dutch surgeons also noted that patients frequently harbored unrealistic expectations of being entirely pain-free following surgery, making it challenging to decrease opioid prescribing.

## Category 3: Collaborative care

Orthopaedic surgeons from Canada and the Netherlands interviewed in this study reported collaborations with other healthcare providers, pain clinics, and pharmaceutical departments to enhance patient care and education, to receive education on opioid prescribing, and to develop a standardized pain management protocol within their institutions (S3 File, S1 and S2 Figs). These collaborations also aimed to provide optimal care for patients with substance use disorders.

Many orthopedic surgeons that we interviewed shared apprehensions about patients obtaining opioid refills from other healthcare providers without their knowledge or authorization. Also, Canadian surgeons expressed challenges with referring patients to pain clinics for more specialized care as these clinics often had long wait times. In contrast, many Japanese surgeons described that their institution lacked a specialized pain management team, making it challenging for them to refer patients with greater needs to professionals who are more experienced in managing pain. Dutch surgeons in our study described inconsistencies in opioid prescribing patterns and attitudes among healthcare professionals within a patient's care team, especially if providers other than the surgeon were involved in prescribing.

## Category 4: Policies, guidelines, and regulations on opioid prescribing

During the interviews, many Canadian surgeons identified provincial campaigns on opioid reduction and institutional regulations and monitoring as facilitators of opioid reduction (S3 File, S1 Fig). Notably, some Canadian surgeons perceived reluctance in the orthopaedic community to prescribe opioids due to fear of litigation or license suspension. Moreover, some Japanese surgeons in our study highlighted that regulations on more potent opioids restricted their ability to prescribe them. However, most Canadian and Japanese surgeons lacked knowledge about existing guidelines for opioid prescribing (S2 Fig). Some Canadian surgeons believed that current policies and guidelines on peri-operative opioid use

were outdated. Some also felt that the strict opioid prescribing policies and restrictions placed pressure on them and hindered their ability to provide individualized care, particularly for patients experiencing severe pain.

### Category 5: Opioid risk mitigation via continuing surgeon education

Included orthopaedic surgeons from all three countries discussed their experiences and perspectives on continuing education regarding safer opioid prescribing (S3 File, S1 and S2 Figs). Some Canadian and Dutch orthopedic surgeons were actively involved in opioid-related research and were members of arthroplasty or orthopedic groups and societies. Others recalled receiving institutional/departmental training on pain management and opioid prescribing (i.e., through modules, meetings, rounds, and conferences). Conversely, while some Japanese institutions provided additional training and guidance on opioid prescribing, many interviewed surgeons reported completing external e-learning modules for every type of opioid that they prescribed.

During interviews, some surgeons in all three countries stated that they had not received any specific education on opioid prescribing after their medical training. While some of these surgeons did not see the need for further education, others recognized its importance. For example, Canadian surgeons emphasized the need for more education on identifying abuse, pain management for vulnerable populations, dosing, non-opioid alternatives, and prescribing trends, with a preference for online and short educational sessions. Likewise, Dutch surgeons in our study identified a need of formal education on available multimodal pain management protocols, relevant literature updates, novel products, and emerging non-opioid alternatives. Japanese surgeons highlighted the need for education on risk factors for substance use disorder and postoperative pain management strategies for patients at risk of substance use disorders.

### Category 6: Personal perceptions and beliefs towards opioid prescribing

Orthopaedic surgeons that we interviewed from Canada, the Netherlands, and Japan had varying perceptions of opioid prescribing, but some similarities were observed (S3 File, S1 Fig). Many Canadian and Dutch surgeons believed that opioids are necessary for improving patients' sleep and mobility and, subsequently, cannot be eliminated from post-THA/TKA pain management protocols. Almost all Japanese orthopaedic surgeons that we interviewed considered tramadol less addictive than other opioids, but some desired more potent and longer-duration opioids. Many included surgeons from Japan and the Netherlands exhibited knowledge gaps and uncertainty regarding the use of opioids for patients with substance use disorders.

## Discussion

We found that all orthopaedic surgeons that we interviewed from Canada and almost all from the Netherlands prescribed opioids to every patient after joint replacement surgery. However, Japanese surgeons in our sample exhibited much greater variability, with half prescribing opioids to all patients and half prescribing to only a minority of patients or none. Consistent with other literature, our findings suggest differences in prescribing practices between Canadian, Dutch, and Japanese orthopaedic surgeons emanate from individual, patient, and system-level disparities [25–28].

Many included orthopaedic surgeons from Japan believed that opioids are largely unnecessary for managing postoperative pain, as opposed to their counterparts from Canada and the Netherlands, who believed that opioids cannot completely be eliminated from postoperative pain management protocols. A survey comparing opioid prescribing patterns among 164 orthopaedic surgeons in the USA and Japan also reported that more participants in the USA strongly believed that opioids are necessary for outpatient post-surgical pain control compared to those in Japan (23% in the USA vs. 3% in Japan) [28]. Likewise, a cross-sectional survey of 981 patients from the USA and seven other countries reported that more American patients were given opioids after discharge compared with those in other countries [27]. All Dutch surgeons that we interviewed used an evidence-based, institutional standard pain protocol and an analgesic ladder, recommending non-opioid therapies as the first line of treatment. In contrast, most surgeons that we included from Canada and Japan did

not describe the use of a standardized protocol for analgesia after surgery. Several studies have reported that the implementation of standard institutional protocols restricting opioid use has resulted in lower mean consumption of opioids, underscoring the need for institutions to establish standardized protocols and ensure adherence [29–31].

Canadian and Dutch surgeons in our study seldom instructed patients on how to dispose of unused opioid medications. Consistent provision of instructions to patients regarding safe storage and disposal methods may serve as a preventive measure against the diversion of unused opioids for potential abuse or misuse [32,33]. Japanese surgeons that we interviewed perceived that instructions on leftover medications were unnecessary. However, a cross-sectional study involving 387 Japanese patients with chronic non-cancer pain revealed that 38% met criteria for opioid misuse and 17% met criteria for opioid abuse [34]. Included surgeons also reported challenges with patients procuring opioid refills from other healthcare providers. A 2018 survey of 2772 US orthopedic trauma patients found that 20% engaged in "doctor shopping" for additional opioids after surgery, highlighting the need for improved communication between primary care and surgeons [35]. Moreover, this underscores the continued importance of educating patients on the risks associated with opioids, expectations of pain management, exploring alternative treatment options, and providing guidance on safe handling and proper disposal methods. A systematic review of 11 studies, evaluating the effectiveness of pre-operative patient education on post-operative opioid use and pain management in the orthopaedic setting, found that education related to opioid use and pain can be effective in reducing opioid prescription requests and filling [36]. Similarly, a cluster-randomized controlled trial including 539 patients reported that patient education on safe opioid disposal approximately triples opioid disposal rates compared with no education [33].

The surgeons that we interviewed from Japan highlighted a social and cultural resistance among their patients to using opioids and, thus, a preference for non-opioid analgesia. This inclination may stem concomitantly from the perception that chronic opioid use may be a criminal offense and cultural mores regarding self-attention, discouraging individuals from voicing pain-related concerns [4]. Moreover, an observational study found that American patients reported higher chronic pain incidence and greater opioid load overall [37]. Japanese surgeons we interviewed that prescribed only opioids administered tramadol, which they perceived as safer than other opioids. However, tramadol's pain-relieving effects depend on opioid metabolites, the quantities of which can differ among individuals due to genetic variations. Consequently, this divergence can lead to varying levels of risk associated with their use. From 2015 to 2017 in the USA, past-year misuse of oral tramadol was approximately 4%, which was lower than that of other opioids [38].

Differences in perceptions of facilitators and barriers between Canada, the Netherlands, and Japan may also stem from variability in healthcare systems. Specifically, Japan has a national insurance system that strictly enforces the indications for each medication, and non-compliance results in a lack of insurance coverage. For instance, the use of oxycodone is only covered by insurance for treating cancer pain and cannot be prescribed for other types of pain [4]. Additionally, Japanese physicians prescribing opioids for chronic non-cancer pain must fulfill several requirements: the physician must have completed e-learning modules for each specific opioid, the physician and patient must sign a contract prior to initiating opioid therapy, the patient must be prescribed non-opioid analgesics first, and the patient must undergo trial use of a challenge opioid to ensure effectiveness [4]. Also, Japanese surgeons included in our study described that patients are hospitalized for several weeks following most elective surgeries, in contrast to the Canadian or Dutch healthcare systems, which are rapidly transitioning towards outpatient care settings. During hospitalization, Japanese patients receive intensive rehabilitation and pain management while closely monitored by medical and nursing staff, resulting in their exposure to opioid medication being subject to strict surveillance [28]. Similarly, many European countries enforce regulatory restrictions on the prescribing of opioids including dose limits, duplicate or triplicate prescriptions using special forms, and strict monitoring by pharmacies [39].

Most surgeons were aware of institutional regulations and restrictions on opioid prescribing, but many surgeons in Canada and the Netherlands lacked knowledge of existing guidelines and policies. The Canadian Orthopedic Association's position statement on opioid prescribing highlights the use of the 2017 Canadian Guideline for Opioids for

Chronic Non-Cancer Pain which offers direction for physicians in Canada regarding the prescription of opioids and advocates for a more conservative approach [40,41]. This guideline is complementary to several province-specific resources and practice standards such as the Safe Prescribing of Opioids and Sedatives practice standard by the College of Physicians and Surgeons of British Columbia and the Public Health Ontario Interactive Opioid Tool [42]. Moreover, the CDC Clinical Practice Guideline for Prescribing Opioids for Pain — United States, 2022, provides recommendations for clinicians prescribing opioids to adult outpatients [43]. Recently, clinical practice recommendations were developed for prescribing opioids to patients with chronic noncancer pain in Europe [44]. However, there is a need for surgery-specific guidelines on opioid prescribing to promote evidence-based practices. Notably, a retrospective cohort study, evaluating postoperative opioid prescription quantities, variability, and 30-day refill rates before and after implementation of orthopedic procedure-specific guidelines reported decreased opioid prescription amounts and variability [45].

Cross-national insights can be integrated into the development of surgery-specific opioid prescribing guidelines. The variability observed across Canada, the Netherlands, and Japan highlights the need for evidence-based protocols that account for both cultural contexts and patient preferences. Our findings suggest that surgeon training programs can integrate updated guidance on multimodal pain management strategies, safe opioid prescribing, and the importance of educating patients on proper storage and disposal of unused opioid medications. Emphasizing existing national and international guidelines within training can further support consistent, informed decision-making. Addressing the identified barriers will require a multifaceted approach, including system-level changes and targeted educational interventions. Future research can evaluate the effectiveness of such strategies in supporting surgeons to adopt patient-centered, evidence-based practices that minimize opioid use while ensuring adequate pain control. Multi-center, quantitative studies are also needed to improve generalizability and explore cultural influences on prescribing behavior. Additionally, qualitative and quantitative studies exploring patient perspectives may offer a more holistic understanding of pain management preferences and inform more tailored interventions.

## Strengths

Our study provides a cross-cultural understanding of opioid prescribing practices by exploring the perspectives of orthopedic surgeons across three countries that represent high (Canada), moderate (the Netherlands), and low (Japan) opioid prescribing practices. The qualitative approach allows for an in-depth exploration of nuanced barriers and facilitators that may not be captured through quantitative methods. Additionally, we implemented triangulation and member checking to enhance the trustworthiness of our findings.

## Limitations

Our study involved surgeons who volunteered to participate and, thus, may have had a heightened interest or stronger perceptions regarding opioid use and prescribing. Also, there were considerably more men than women participants in our study, likely reflecting the under-representation of women in arthroplasty in all three countries of interest [46]. Our results may not be transferable to other settings due to differences in population makeup and cultures. Moreover, our sample primarily consisted of surgeons practicing in academic settings, which may stem from inherent limitations in convenience and snowball sampling. As qualitative studies seek to understand a specific phenomenon, generalizability is not an expected attribute as it is for quantitative studies. Additionally, interviews were conducted in English, which may have created comprehension difficulties for some Japanese surgeons. However, we provided the option of a translator if necessary. Although there is a potential for social desirability bias in participants' responses, we endeavored to minimize this bias by ensuring that there were no leading questions, using indirect questioning, probing and clarifying participant responses, and providing assurance of confidentiality and anonymity prior to commencing the interviews [47].

## Conclusions

Canadian, Dutch, and Japanese orthopedic surgeons in our study delineated a range of individual, patient, and system level contributors to variability in opioid prescribing after joint replacement surgery. These findings suggest that multifactorial and context-specific approaches may be required to address barriers and optimize postoperative use of opioids. Rigorous quantitative studies, with larger sample sizes, are needed to validate these qualitative findings and assess their generalizability across broader populations. Additionally, further qualitative and quantitative studies, exploring patient perspectives on opioid use, are needed for achieving a more comprehensive understanding of the issue.

## Supporting information

**S1 File. Semi-Structured Interview Guide (30–45 Minutes).**
(DOCX)

**S2 File. Interviewer Positionality Statements.**
(DOCX)

**S3 File.** S3 Table 1. Pain medications prescribed by orthopaedic surgeons in Canada, Japan, and the Netherlands. S3 Table 2. Facilitators and barriers to reduced opioid prescribing influencing surgeons' prescribing practises. S3 Table 3. Facilitators and barriers to reduced opioid prescribing related to patient factors/perspectives. S3 Table 4. Facilitators and barriers to reduced opioid prescribing related to providing collaborative patient care. S3 Table 5. Facilitators and barriers to reduced opioid prescribing related to policies, guidelines, and regulations on opioid prescribing. S3 Table 6. Barriers and facilitators to reduced opioid prescribing related to opioid risk mitigation via surgeon education. S3 Table 7. Surgeons' personal perceptions and beliefs on opioid prescribing.
(DOCX)

**S1 Fig. Comparison of factors perceived by surgeons as facilitating opioid reduction in Canada, Japan, and the Netherlands.**
(TIFF)

**S2 Fig. Comparison of factors perceived by surgeons as hindering opioid reduction in Canada, Japan, and the Netherlands.**
(TIFF)

**S4 File. Inclusivity in global research.**
(DOCX)

## Author contributions

**Conceptualization:** Mansi Patel, Kim Madden, Sheila Sprague, Jason W. Busse.

**Data curation:** Mansi Patel, Parsia Parnian, Natsumi Saka, Jason W. Busse.

**Formal analysis:** Mansi Patel, Parsia Parnian.

**Funding acquisition:** Mansi Patel, Jason W. Busse.

**Investigation:** Mansi Patel, Parsia Parnian, Kim Madden, Anita Acai, Ydo Kleinlugtenbelt, Natsumi Saka, Ellie Landman, Vickas Khanna, Jason W. Busse.

**Methodology:** Mansi Patel, Kim Madden, Sheila Sprague, Anita Acai, Ydo Kleinlugtenbelt, Ellie Landman, Harsha Shanthanna, Vickas Khanna, Jason W. Busse.

**Project administration:** Kim Madden, Sheila Sprague, Anita Acai, Ydo Kleinlugtenbelt, Ellie Landman.

**Resources:** Kim Madden, Sheila Sprague, Anita Acai, Ydo Kleinlugtenbelt, Natsumi Saka, Ellie Landman, Harsha Shanthanna, Vickas Khanna, Jason W. Busse.

**Software:** Mansi Patel, Parsia Parnian.

**Supervision:** Kim Madden, Sheila Sprague, Anita Acai, Ydo Kleinlugtenbelt, Ellie Landman, Jason W. Busse.

**Visualization:** Mansi Patel.

**Writing – original draft:** Mansi Patel, Parsia Parnian.

**Writing – review & editing:** Mansi Patel, Parsia Parnian, Kim Madden, Sheila Sprague, Anita Acai, Ydo Kleinlugtenbelt, Natsumi Saka, Ellie Landman, Harsha Shanthanna, Vickas Khanna, Jason W. Busse.

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
