## [Decision Letter · Decision Letter 0]

15 Dec 2024

PONE-D-24-31327Perceptions of barriers and facilitators to opioid reduction after total joint arthroplasty among orthopedic surgeons practicing in Canada, Japan, and the Netherlands: A qualitative description studyPLOS ONE

Dear Dr. Busse,

Thank you for submitting your manuscript to PLOS ONE. After careful consideration, we feel that it has merit but does not fully meet PLOS ONE’s publication criteria as it currently stands. Therefore, we invite you to submit a revised version of the manuscript that addresses the points raised during the review process.

We look forward to receiving your revised manuscript.

Kind regards,

Sina Afzal

Academic Editor

PLOS ONE

Journal Requirements:

b). In the Methods ethics statement, you specified that verbal consent was obtained. Please provide additional details regarding how this consent was documented and witnessed, and state whether this was approved by the IRB.

3. Thank you for stating the following in the Competing Interests section: [I have read the journal's policy and the authors of this manuscript have the following competing interests:

One of the authors (MP) has received funding from the Mitacs Globalink Research Award

One of the authors (VK) is a consultant for Stryker Canada and Zimmer Biomet Canada, and has a spouse who works for Stryker Canada.

One of the authors (YK) received a grant from Pioneers in Healthcare, received honoraria from Agnovos, received support for meeting attendance from RGS, and payment from NOV for participation in a Data Safety Monitoring Board or Advisory Board.

One of the authors (KM) received a grant from CIHR (BioTalent), is a participant in a Data Safety Monitoring Board or Advisory Board at the University of Calgary, and has a leadership or fiduciary role at the Clinical Orthopaedics and Related Research and Canadian Orthopaedic Association.

All other authors (PP, SS, AA, NS, EL, HS, JWB) certify that there are no funding or commercial associations (consultancies, stock ownership, equity interest, patent/licensing arrangements, etc.) that might pose a conflict of interest in connection with the submitted article related to the author or any immediate family members.]. Please confirm that this does not alter your adherence to all PLOS ONE policies on sharing data and materials, by including the following statement: "This does not alter our adherence to PLOS ONE policies on sharing data and materials.” (as detailed online in our guide for authors http://journals.plos.org/plosone/s/competing-interests). If there are restrictions on sharing of data and/or materials, please state these. Please note that we cannot proceed with consideration of your article until this information has been declared. Please include your updated Competing Interests statement in your cover letter; we will change the online submission form on your behalf.

4. In this instance it seems there may be acceptable restrictions in place that prevent the public sharing of your minimal data. However, in line with our goal of ensuring long-term data availability to all interested researchers, PLOS’ Data Policy states that authors cannot be the sole named individuals responsible for ensuring data access (http://journals.plos.org/plosone/s/data-availability#loc-acceptable-data-sharing-methods). Data requests to a non-author institutional point of contact, such as a data access or ethics committee, helps guarantee long term stability and availability of data. Providing interested researchers with a durable point of contact ensures data will be accessible even if an author changes email addresses, institutions, or becomes unavailable to answer requests. Before we proceed with your manuscript, please also provide non-author contact information (phone/email/hyperlink) for a data access committee, ethics committee, or other institutional body to which data requests may be sent. If no institutional body is available to respond to requests for your minimal data, please consider if there any institutional representatives who did not collaborate in the study, and are not listed as authors on the manuscript, who would be able to hold the data and respond to external requests for data access? If so, please provide their contact information (i.e., email address). Please also provide details on how you will ensure persistent or long-term data storage and availability.

5. Please upload a copy of Figures 1 and 2, to which you refer in your text on page 10. If the figure is no longer to be included as part of the submission please remove all reference to it within the text.

Reviewers' comments:

Reviewer's Responses to Questions

**Comments to the Author**

1. Is the manuscript technically sound, and do the data support the conclusions?

Reviewer #1: Yes

Reviewer #2: Yes

2. Has the statistical analysis been performed appropriately and rigorously? 

Reviewer #1: Yes

Reviewer #2: I Don't Know

3. Have the authors made all data underlying the findings in their manuscript fully available?

Reviewer #1: Yes

Reviewer #2: No

4. Is the manuscript presented in an intelligible fashion and written in standard English?

Reviewer #1: Yes

Reviewer #2: Yes

5. Review Comments to the Author

Reviewer #1: This article addresses a critical and timely issue of reducing opioid use following joint arthroplasty, providing valuable insights through a cross-cultural comparison of prescribing practices in different countries.

The article could be improved with the following suggestions:

- The results compare countries but lack sufficient emphasis on the unique cultural and systemic factors influencing opioid prescribing behaviors. The suggestion of providing more detailed comparisons between countries focuses on how cultural and systemic differences shape prescribing practices. Including a table or figure summarizing these differences would improve clarity.

-The discussion could benefit from stronger connections to existing literature, particularly around the role of national policies and patient education in opioid prescribing.

- While the study identifies barriers, it lacks concrete recommendations for overcoming them. Suggestions for specific, actionable strategies tailored to each country, such as surgeon training programs, standardized protocols, or policy initiatives.

- Limitations Section: The authors acknowledge some limitations (e.g., gender imbalance), but others are understated, such as potential biases arising from interviewing surgeons who may already hold strong views on opioid prescribing.

- Further suggestion is to address the strengths of this study.

- Additional Recommendations

--Incorporate Practical Insights:

Highlight how the findings can directly inform policies or training programs for surgeons.

--Ethical Considerations:

Elaborate on measures to ensure confidentiality and address potential power dynamics during interviews, which could influence participants' responses.

--Future Research Directions:

Discuss the need for follow-up quantitative studies to validate the qualitative findings and explore patient perspectives to provide a more comprehensive understanding of the issue.

Reviewer #2: I would like to thank all the authors for their efforts in conducting this quality study. Nearly all the key issues and limitations associated with this research have been thoughtfully acknowledged and addressed within the manuscript itself.

6. PLOS authors have the option to publish the peer review history of their article (what does this mean? ). If published, this will include your full peer review and any attached files.

**Do you want your identity to be public for this peer review?** For information about this choice, including consent withdrawal, please see our Privacy Policy .

Reviewer #1: No

Reviewer #2: No

---

## [Author Response · Author response to Decision Letter 1]

27 Jan 2025

RE: REVISION: Perceptions of barriers and facilitators to opioid reduction after total joint arthroplasty among orthopedic surgeons practicing in Canada, Japan, and the Netherlands: A qualitative description study

Dear Dr. Chenette,

Please accept our revisions for the manuscript submission entitled, “Perceptions of barriers and facilitators to opioid reduction after total joint arthroplasty among orthopedic surgeons practicing in Canada, Japan, and the Netherlands: A qualitative description study” for consideration for publication in PLOS One.

We would like to thank the reviewers and editors for their time and effort in reviewing our manuscript and providing comments. We appreciate the quality of the feedback and have edited our manuscript to align with your comments as much as possible, providing justifications where required.

Journal Requirements:

Thank you, we have re-checked the formatting to ensure it aligns with the requirements.

Thank you for this information. We have completed this questionnaire and have submitted it as a supplementary information file (S4 File). We referred to this file on lines 174-176.

a. In the Methods ethics statement, you specified that verbal consent was obtained. Please provide additional details regarding how this consent was documented and witnessed, and state whether this was approved by the IRB.

Thank you. We have added further details in the Methods section, lines 131-137, “Prior to data collection, participants were provided with a research ethics board (REB)-approved information letter that described the study in accessible language and provided sufficient information for participants to make an informed decision about their participation. They were given ample time to review and ask questions. Thereupon, verbal informed consent was obtained during the interview and documented on a REB-approved verbal informed consent log for audio and/or video recording and the use of quotes.”

For further clarity, all participants included in this study reviewed a REB-approved information letter before data collection that described this study in accessible language and provided sufficient information for participants to make an informed decision about their participation. The information letter described the study purpose, detailed background information, the nature of participation, and associated risks and benefits in lay language. Participants were also provided with the contact information of the study investigators, the principal investigator(s), and individual(s) outside the research team who can be contacted regarding questions or possible ethical issues. Then, verbal informed consent was obtained during the interview and documented on a verbal informed consent log, approved by the local IRB.

3. Thank you for stating the following in the Competing Interests section: [I have read the journal's policy and the authors of this manuscript have the following competing interests:

One of the authors (MP) has received funding from the Mitacs Globalink Research Award

One of the authors (VK) is a consultant for Stryker Canada and Zimmer Biomet Canada, and has a spouse who works for Stryker Canada.

One of the authors (YK) received a grant from Pioneers in Healthcare, received honoraria from Agnovos, received support for meeting attendance from RGS, and payment from NOV for participation in a Data Safety Monitoring Board or Advisory Board.

One of the authors (KM) received a grant from CIHR (BioTalent), is a participant in a Data Safety Monitoring Board or Advisory Board at the University of Calgary, and has a leadership or fiduciary role at the Clinical Orthopaedics and Related Research and Canadian Orthopaedic Association.

All other authors (PP, SS, AA, NS, EL, HS, JWB) certify that there are no funding or commercial associations (consultancies, stock ownership, equity interest, patent/licensing arrangements, etc.) that might pose a conflict of interest in connection with the submitted article related to the author or any immediate family members.].

a. Please confirm that this does not alter your adherence to all PLOS ONE policies on sharing data and materials, by including the following statement: "This does not alter our adherence to PLOS ONE policies on sharing data and materials.” (as detailed online in our guide for authors http://journals.plos.org/plosone/s/competing-interests). If there are restrictions on sharing of data and/or materials, please state these. Please note that we cannot proceed with consideration of your article until this information has been declared. Please include your updated Competing Interests statement in your cover letter; we will change the online submission form on your behalf.

Thank you for letting us know. We have included the Competing Interests statement along with the aforementioned statement in our revised cover letter.

“Please note the following competing interests: One of the authors (MP) has received funding from the Mitacs Globalink Research Award. One of the authors (VK) is a consultant for Stryker Canada and Zimmer Biomet Canada, and has a spouse who works for Stryker Canada. One of the authors (YK) received a grant from Pioneers in Healthcare, received honoraria from Agnovos, received support for meeting attendance from RGS, and payment from NOV for participation in a Data Safety Monitoring Board or Advisory Board. One of the authors (KM) received a grant from CIHR (BioTalent), is a participant in a Data Safety Monitoring Board or Advisory Board at the University of Calgary, and has a leadership or fiduciary role at the Clinical Orthopaedics and Related Research and Canadian Orthopaedic Association. All other authors (PP, SS, AA, NS, EL, HS, JWB) certify that there are no funding or commercial associations (consultancies, stock ownership, equity interest, patent/licensing arrangements, etc.) that might pose a conflict of interest in connection with the submitted article related to the author or any immediate family members. This does not alter our adherence to PLOS ONE policies on sharing data and materials.”

4. In this instance it seems there may be acceptable restrictions in place that prevent the public sharing of your minimal data. However, in line with our goal of ensuring long-term data availability to all interested researchers, PLOS’ Data Policy states that authors cannot be the sole named individuals responsible for ensuring data access (http://journals.plos.org/plosone/s/data-availability#loc-acceptable-data-sharing-methods). Data requests to a non-author institutional point of contact, such as a data access or ethics committee, helps guarantee long term stability and availability of data. Providing interested researchers with a durable point of contact ensures data will be accessible even if an author changes email addresses, institutions, or becomes unavailable to answer requests. Before we proceed with your manuscript, please also provide non-author contact information (phone/email/hyperlink) for a data access committee, ethics committee, or other institutional body to which data requests may be sent. If no institutional body is available to respond to requests for your minimal data, please consider if there any institutional representatives who did not collaborate in the study, and are not listed as authors on the manuscript, who would be able to hold the data and respond to external requests for data access? If so, please provide their contact information (i.e., email address). Please also provide details on how you will ensure persistent or long-term data storage and availability.

Thank you. We have submitted all data to MacSphere, McMaster University's open-access institutional repository. Upon approval, this will ensure long-term storage and accessibility of the data. For any data requests, please contact the McMaster University Office of Scholarly Communication at scom@mcmaster.ca.

5. Please upload a copy of Figures 1 and 2, to which you refer in your text on page 10. If the figure is no longer to be included as part of the submission please remove all reference to it within the text.

Thank you for letting us know. We have uploaded S3 Fig 1 and S3 Fig 2 (included in S3 File) separately as well.

We have reviewed the reference list and ensure that it is complete and correct.

Reviewers' Comments:

1. Is the manuscript technically sound, and do the data support the conclusions?

a. Reviewer #1: Yes

b. Reviewer #2: Yes

Thank you.

2. Has the statistical analysis been performed appropriately and rigorously?

a. Reviewer #1: Yes

b. Reviewer #2: I Don't Know

Thank you. Due to the qualitative nature of this study, data analysis involved conventional (inductive) content analysis. We summarized demographic data using ranges and proportions.

3. Have the authors made all data underlying the findings in their manuscript fully available? The PLOS Data policy requires authors to make all data underlying the findings described in their manuscript fully available without restriction, with rare exception (please refer to the Data Availability Statement in the manuscript PDF file). The data should be provided as part of the manuscript or its supporting information, or deposited to a public repository. For example, in addition to summary statistics, the data points behind means, medians and variance measures should be available. If there are restrictions on publicly sharing data—e.g. participant privacy or use of data from a third party—those must be specified.

a. Reviewer #1: Yes

b. Reviewer #2: No

To protect the privacy or confidentiality of human research participants, not all data can be made available publicly (i.e. direct quotations of participants that may reveal their identity). Interested researchers can contact the corresponding author or McMaster University Office of Scholarly Communication at scom@mcmaster.ca to access the minimal data that we have. They can also access this data at https://macsphere.mcmaster.ca.

4. Is the manuscript presented in an intelligible fashion and written in standard English? PLOS ONE does not copyedit accepted manuscripts, so the language in submitted articles must be clear, correct, and unambiguous. Any typographical or grammatical errors should be corrected at revision, so please note any specific errors here.

a. Reviewer #1: Yes

b. Reviewer #2: Yes

Thank you.

Review Comments to the Author:

5. Reviewer #1: This article addresses a critical and timely issue of reducing opioid use following joint arthroplasty, providing valuable insights through a cross-cultural comparison of prescribing practices in different countries.

Thank you for your thorough feedback and taking the time to review our manuscript.

The article could be improved with the following suggestions:

a. The results compare countries but lack sufficient emphasis on the unique cultural and systemic factors influencing opioid prescribing behaviors. The suggestion of providing more detailed comparisons between countries focuses on how cultural and systemic differences shape prescribing practices. Including a table or figure summarizing these differences would improve clarity.

The results summarize the perceptions of orthopaedic surgeons that we identified through the interview sessions. Considering the qualitative description methodology used in the study, we provided an analytical and data-driven perspective, by using conventional (inductive) content analysis, rather than interpreting the findings. We highlighted differences and similarities in the perceptions of orthopaedic surgeons identified during interviews from each of the 3 countries in S3 Tables 2 to 7 and S3 Fig 1 and S3 Fig 2.

b. The discussion could benefit from stronger connections to existing literature, particularly around the role of national policies and patient education in opioid prescribing.

Thank you for your feedback. We have discussed the role of national policies in opioid prescribing in Japan (Lines 374-387). We have also added the following to the discussion:

- “Notably, a retrospective cohort study, evaluating postoperative opioid prescription quantities, variability, and 30-day refill rates before and after implementation of orthopedic procedure-specific guidelines reported decreased opioid prescription amounts and variability” (Lines 406-409).

- “Moreover, this underscores the continued importance of educating patients on the risks associated with opioids, expectations of pain management, exploring alternative treatment options, and providing guidance on safe handling and proper disposal methods. A systematic review of 11 studies, evaluating the effectiveness of pre-operative patient education on post-operative opioid use and pain management in the orthopaedic setting, found that education related to opioid use and pain can be effective in reducing opioid prescription requests and filling[36]. Similarly, a cluster-randomized controlled trial including 539 patients reported that patient education on safe opioid disposal approximately triples opioid disposal rates compared with no education [33].” (Lines 349-358).

- “Similarly, many European countries enforce regulatory restrictions on the prescribing of opioids including dose limits, duplicate or triplicate prescriptions using special forms, and strict monitoring by pharmacies [39]” (Lines 387-389).

c. While the study identifies barriers, it lacks concrete recommendations for overcoming them. Suggestions for specific, actionable strategies tailored to each country, such as surgeon training programs, standardized protocols, or policy initiatives.

The scope of this qualitative study is limited as it serves as a preliminary exploration with a small sample size, and our findings may not gen

---

## [Decision Letter · Decision Letter 1]

18 Jun 2025

PONE-D-24-31327R1Perceptions of barriers and facilitators to opioid reduction after total joint arthroplasty among orthopedic surgeons practicing in Canada, Japan, and the Netherlands: A qualitative description studyPLOS ONE

Dear Dr. Busse,

Thank you for submitting your manuscript to PLOS ONE. After careful consideration, we feel that it has merit but does not fully meet PLOS ONE’s publication criteria as it currently stands. Therefore, we invite you to submit a revised version of the manuscript that addresses the points raised during the review process.

We look forward to receiving your revised manuscript.

Kind regards,

Miquel Vall-llosera Camps

Senior Staff Editor

PLOS One

Journal Requirements:

Reviewers' comments:

Reviewer's Responses to Questions

**Comments to the Author**

1. If the authors have adequately addressed your comments raised in a previous round of review and you feel that this manuscript is now acceptable for publication, you may indicate that here to bypass the “Comments to the Author” section, enter your conflict of interest statement in the “Confidential to Editor” section, and submit your "Accept" recommendation.

Reviewer #1: All comments have been addressed

2. Is the manuscript technically sound, and do the data support the conclusions?

Reviewer #1: Yes

3. Has the statistical analysis been performed appropriately and rigorously? 

Reviewer #1: Yes

4. Have the authors made all data underlying the findings in their manuscript fully available?

Reviewer #1: Yes

5. Is the manuscript presented in an intelligible fashion and written in standard English?

Reviewer #1: Yes

6. Review Comments to the Author

Reviewer #1: Dear Authors,

Thank you for thoroughly addressing the previous review comments. Your revisions have significantly improved the manuscript, particularly in the areas of cultural context, policy implications, and study limitations. I appreciate your efforts in providing additional data and strengthening the discussion section.

Final Recommendations:

Practical Application:

While you have acknowledged the preliminary nature of this study, including a brief section on how these findings could inform clinical guidelines or surgeon training would enhance the manuscript’s real-world impact.

Future Research Directions:

Your inclusion of future research needs is appreciated. However, a more explicit mention of potential multi-center studies or patient perspectives could further guide subsequent investigations in this field.

Overall, the manuscript now meets the scientific and editorial standards for publication, and I recommend acceptance .

Thank you for your dedication to advancing this important area of research.

7. PLOS authors have the option to publish the peer review history of their article (what does this mean? ). If published, this will include your full peer review and any attached files.

**Do you want your identity to be public for this peer review?** For information about this choice, including consent withdrawal, please see our Privacy Policy .

Reviewer #1: No

---

## [Author Response · Author response to Decision Letter 2]

18 Jun 2025

Dear Dr. Chenette,

Please accept our revisions for the manuscript submission entitled, “Perceptions of barriers and facilitators to opioid reduction after total joint arthroplasty among orthopedic surgeons practicing in Canada, Japan, and the Netherlands: A qualitative description study” for consideration for publication in PLOS One.

We would like to thank the reviewers and editors for their time and effort in reviewing our manuscript and providing comments. We appreciate the quality of the feedback and have edited our manuscript to align with your comments as much as possible, providing justifications where required.

Journal Requirements:

Please review your reference list to ensure that it is complete and correct. If you have cited papers that have been retracted, please include the rationale for doing so in the manuscript text or remove these references and replace them with relevant current references. Any changes to the reference list should be mentioned in the rebuttal letter that accompanies your revised manuscript. If you need to cite a retracted article, indicate the article’s retracted status in the References list and also include a citation and full reference for the retraction notice.

We have checked the reference list and ensure that it is up-to-date and complete.

Reviewers' comments:

1. If the authors have adequately addressed your comments raised in a previous round of review and you feel that this manuscript is now acceptable for publication, you may indicate that here to bypass the “Comments to the Author” section, enter your conflict-of-interest statement in the “Confidential to Editor” section, and submit your "Accept" recommendation.

Reviewer #1: All comments have been addressed

Thank you.

2. Is the manuscript technically sound, and do the data support the conclusions?

Reviewer #1: Yes

Thank you.

3. Has the statistical analysis been performed appropriately and rigorously?

Reviewer #1: Yes

Thank you.

4. Have the authors made all data underlying the findings in their manuscript fully available?

Reviewer #1: Yes

Thank you.

5. Is the manuscript presented in an intelligible fashion and written in standard English?

Reviewer #1: Yes

Thank you.

6. Review Comments to the Author

Reviewer #1:

Dear Authors,

Thank you for thoroughly addressing the previous review comments. Your revisions have significantly improved the manuscript, particularly in the areas of cultural context, policy implications, and study limitations. I appreciate your efforts in providing additional data and strengthening the discussion section.

Thank you for your feedback.

Final Recommendations:

Practical Application:

While you have acknowledged the preliminary nature of this study, including a brief section on how these findings could inform clinical guidelines or surgeon training would enhance the manuscript’s real-world impact.

Thank you for your feedback. We have added the following section in the Discussion:

“Cross-national insights can be integrated into the development of surgery-specific opioid prescribing guidelines. The variability observed across Canada, the Netherlands, and Japan highlights the need for evidence-based protocols that account for both cultural contexts and patient preferences. Our findings suggest that surgeon training programs can integrate updated guidance on multimodal pain management strategies, safe opioid prescribing, and the importance of educating patients on proper storage and disposal of unused opioid medications. Emphasizing existing national and international guidelines within training can further support consistent, informed decision-making. Addressing the identified barriers will require a multifaceted approach, including system-level changes and targeted educational interventions. Future research can evaluate the effectiveness of such strategies in supporting surgeons to adopt patient-centered, evidence-based practices that minimize opioid use while ensuring adequate pain control. Multi-center, quantitative studies are also needed to improve generalizability and explore cultural influences on prescribing behavior. Additionally, qualitative and quantitative studies exploring patient perspectives may offer a more holistic understanding of pain management preferences and inform more tailored interventions.” (Lines 404-418).

Future Research Directions:

Your inclusion of future research needs is appreciated. However, a more explicit mention of potential multi-center studies or patient perspectives could further guide subsequent investigations in this field.

Thank you for your feedback. We have added the following section in the Discussion:

“Cross-national insights can be integrated into the development of surgery-specific opioid prescribing guidelines. The variability observed across Canada, the Netherlands, and Japan highlights the need for evidence-based protocols that account for both cultural contexts and patient preferences. Our findings suggest that surgeon training programs can integrate updated guidance on multimodal pain management strategies, safe opioid prescribing, and the importance of educating patients on proper storage and disposal of unused opioid medications. Emphasizing existing national and international guidelines within training can further support consistent, informed decision-making. Addressing the identified barriers will require a multifaceted approach, including system-level changes and targeted educational interventions. Future research can evaluate the effectiveness of such strategies in supporting surgeons to adopt patient-centered, evidence-based practices that minimize opioid use while ensuring adequate pain control. Multi-center, quantitative studies are also needed to improve generalizability and explore cultural influences on prescribing behavior. Additionally, qualitative and quantitative studies exploring patient perspectives may offer a more holistic understanding of pain management preferences and inform more tailored interventions.” (Lines 404-418).

Overall, the manuscript now meets the scientific and editorial standards for publication, and I recommend acceptance.

Thank you for your dedication to advancing this important area of research.

Thank you for your feedback.

Thank you.

---

## [Editor Report · Decision Letter 2]

14 Aug 2025

Perceptions of barriers and facilitators to opioid reduction after total joint arthroplasty among orthopedic surgeons practicing in Canada, Japan, and the Netherlands: A qualitative description study

PONE-D-24-31327R2

Dear Dr. Busse,

We’re pleased to inform you that your manuscript has been judged scientifically suitable for publication and will be formally accepted for publication once it meets all outstanding technical requirements.

Kind regards,

Miquel Vall-llosera Camps

Senior Staff Editor

PLOS ONE
---

## [Editor Report · Acceptance letter]

PONE-D-24-31327R2

PLOS ONE

Dear Dr. Busse,

I'm pleased to inform you that your manuscript has been deemed suitable for publication in PLOS ONE. Congratulations! Your manuscript is now being handed over to our production team.

Kind regards,

on behalf of

Dr. Miquel Vall-llosera Camps

Staff Editor

PLOS ONE